# Examining the Relationship between Semiquantitative Methods Analysing Concentration-Time and Enhancement-Time Curves from Dynamic-Contrast Enhanced Magnetic Resonance Imaging and Cerebrovascular Dysfunction in Small Vessel Disease [note 1]

**DOI:** 10.3390/jimaging6060043

**Published:** 2020-06-05

**Authors:** Jose Bernal, María Valdés-Hernández, Javier Escudero, Eleni Sakka, Paul A. Armitage, Stephen Makin, Rhian M. Touyz, Joanna M. Wardlaw

**Affiliations:** 1Centre for Clinical Brain Sciences, University of Edinburgh, Edinburgh EH16 4SB, UK; eleni.sakka@ed.ac.uk (E.S.); Joanna.Wardlaw@ed.ac.uk (J.M.W.); 2School of Engineering, University of Edinburgh, Edinburgh EH9 3FB, UK; javier.escudero@ed.ac.uk; 3Academic Unit of Radiology, University of Sheffield, Sheffield S10 2JF, UK; p.armitage@sheffield.ac.uk; 4Institute of Cardiovascular and Medical Sciences, University of Glasgow, Glasgow G12 8TA, UK; Stephen.Makin@glasgow.ac.uk (S.M.); Rhian.Touyz@glasgow.ac.uk (R.M.T.)

**Keywords:** cerebrovascular alteration descriptors, small vessel disease, dynamic contrast-enhanced magnetic resonance imaging

## Abstract

Dynamic contrast-enhanced magnetic resonance imaging (DCE-MRI) can be used to examine the distribution of an intravenous contrast agent within the brain. Computational methods have been devised to analyse the contrast uptake/washout over time as reflections of cerebrovascular dysfunction. However, there have been few direct comparisons of their relative strengths and weaknesses. In this paper, we compare five semiquantitative methods comprising the slope and area under the enhancement-time curve, the slope and area under the concentration-time curve ($Slope_{Con}$ and $AUC_{Con}$), and changes in the power spectrum over time. We studied them in cerebrospinal fluid, normal tissues, stroke lesions, and white matter hyperintensities (WMH) using DCE-MRI scans from a cohort of patients with small vessel disease (SVD) who presented mild stroke. The total SVD score was associated with $AUC_{Con}$ in WMH (p<0.05), but not with the other four methods. In WMH, we found higher $AUC_{Con}$ was associated with younger age (p<0.001) and fewer WMH (p<0.001), whereas $Slope_{Con}$ increased with younger age (p>0.05) and WMH burden (p>0.05). Our results show the potential of different measures extracted from concentration-time curves extracted from the same DCE examination to demonstrate cerebrovascular dysfunction better than those extracted from enhancement-time curves.

## 1. Introduction

Dynamic contrast-enhanced magnetic resonance imaging (DCE-MRI) in the brain is typically considered for examining the integrity of grey and white matter and potential contrast leakage into cerebrospinal fluid (CSF) cavities [1,2,3]. In this imaging modality, a series of MRI scans are taken before and after intravenous Gadolinium-based contrast agent administration to image signal-time trajectories of both healthy and pathological brain regions, as illustrated in Figure 1. Such trajectories may vary depending on the capillary density and the disruption of the blood–brain barrier or blood–CSF barrier, among other factors. Therefore, their precise analysis may help to understand better the mechanisms behind pathological cerebrovascular processes.

Computational approximations for studying signal-time trajectories are classified into two groups: semiquantitative and quantitative [3]. The former group of methods does not make any assumptions about the distribution of contrast agent within the brain, i.e., are model free. Methods that fall under this category analyse the area under the enhancement curve [4], signal enhancement slope [5,6], and dynamic spectral and texture features [7,8]. The latter group of methods describes signal-time curves as a result of interactions between cerebral capillaries and the extracellular extravascular space through pharmacokinetic modelling. The approximation consists of estimating unknown model parameter values from the input data through regression. However, factors such as scanner signal drift, tissue variations, and imaging artefacts introduce systematic errors hampering quantitative assessments [3,9,10,11].

In this work, we study to what extent semiquantitative methods analysing signal-time trajectories from the same imaging acquisitions reflect cerebrovascular dysfunction. In particular, we examine the strength of their association with clinical parameters: the higher the percentage of variance explained by clinical variables, the more the relevant information the measurement captures. The considered methods are (1) the area under the enhancement-time curve, (2) the slope of the enhancement-time curve, (3) the area under the concentration-time curve, (4) the slope of the concentration-time curve, and (5) the radial power spectrum of the concentration-time curve. We use data from a relatively large cohort (n=201) of patients who had a mild stroke and present a varied range of small vessel disease (SVD) features. The main finding of our work is that the analysis of concentration-time curves reflect key aspects of cerebrovascular dysfunction better than enhancement-time curves.

## 2. Materials and Methods

The processing pipeline consists of four steps, as illustrated in Figure 2. First, we acquire structural and dynamic scans for each patient in the cohort. Second, we segment all regions of interest. Third, we analyse contrast-time trajectories in each region. Fourth, we use ANOVA and multiple linear regression to establish whether measurements of contrast uptake/washout vary with any of the clinical variables. Further details of each step are provided in the following sections.

### 2.1. Subjects, Imaging, and Clinical Variables

We used DCE-MRI and clinical data from 201 mild stroke patients with various extents of neuroimaging features of SVD [12,13]. The study was approved by the Lothian Ethics of Medical Research Committee (REC 09/81101/54) and the NHS Lothian R+D Office (2009/W/NEU/14), and all patients gave written informed consent. DCE-MRI scans were obtained approximately a month after stroke presentation using a 3D T1-w spoiled gradient echo sequence (TR/TE = 8.24/3.1 ms, 12${}^{\circ }$ flip angle, 24 cm × 24 cm FoV, 0.9375 mm × 1.25 mm × 4 mm acquired resolution). Following a pre-contrast scan, patients were scanned after an intravenous bolus injection of 0.1mmol/kg of gadoterate meglumine (Gd-DOTA, Dotarem, Guerbet, France) every 73 s during 25 min (leading to 21 frames). We considered age, biological sex (60% male, 40% female), smoker (ever smoker 65% vs. never smoker 35%), diabetes (yes 12% vs. no 88%), hyperlipidaemia (yes 60% vs. no 40%), mean arterial pressure, the total Fazekas score [14] (0–3%, 1–8%, 2–37%, 3–11%, 4–15%, 5–10%, and 6–16%) and the total SVD score [15] (0–33%, 1–24%, 2–23%, 3–13%, and 4–7%).

### 2.2. Segmentation of Regions of Interest

We examined five regions of interest comprising cerebrospinal fluid, deep grey matter, normal-appearing white matter, WMH, and stroke lesions. To obtain their segment masks, we followed the protocol described in [12], i.e., initial segmentation using validated methods, manual edit by trained analysts, and mask erosion to avoid partial volume [11]. Analysts carried out the rectification process blinded to any other imaging and patient information. Details regarding the validation of the segmentation method and inter-analyst agreement can be found in [7,11,12,16].

### 2.3. Methods

#### 2.3.1. Methods Analysing the Enhancement-Time Trajectory

Let S[t]∈R, t∈{0,…,T−1} be the measured signal over time, where S[0] represents the signal value before intravenous contrast injection and *T* the number of time points. Methods analysing the enhancement-time trajectory examine relative signal changes before and after contrast, i.e., (S[t]−S[0])/S[0]. We estimated the area under the enhancement-time curve,
(1)$$AUC_{Enh}=\sum_{t=0}^{T-1}\frac{S[t]-S[0]}{S[0]},$$
and its slope,
(2)$$Slope_{Enh}=\frac{\left(T-t^{*}\right)\cdot \sum_{t=t^{*}}^{T-1}t\cdot S[t]-\sum_{t=t^{*}}^{T-1}t\cdot \sum_{t=t^{*}}^{T-1}S[t]}{\left(T-t^{*}\right)\cdot \sum_{t=t^{*}}^{T-1}t^{2}-{\left(\sum_{t=t^{*}}^{T-1}t\right)}^{2}},$$
where $t^{*}$ is the time point from where the signal trend is assumed linear. Note that the formulation of the slope permits avoiding the peak of bolus arrival. In our case, we set $t^{*}=4$ after visual inspection of all cases in the cohort.

#### 2.3.2. Methods Analysing the Concentration-Time Trajectory

Signal information and imaging parameters can be used to approximate the contrast agent concentration [in millimoles] in time in each region of interest [9]. For that, we converted signal-time curves to concentration-time curves by finding, c[t], that minimises the following expression
(3)$$\min_{c[t]}{∥\frac{S[t]-S[0]}{S[0]}-e^{-r_{2}c\left[t\right]TE}\frac{1-e^{-P-Q[t]}-\cos\left(\theta_{FA}\right)\left(e^{-P}-e^{-2P-Q[t]}\right)}{1-e^{-P}-\cos\left(\theta_{FA}\right)\left(e^{-P-Q[t]}-e^{-2P-Q[t]}\right)}∥}_{2}^{2},$$
where P=TR/T1[0], $Q\left[t\right]=r_{1}\cdot c\left[t\right]\cdot TR$; $r_{1}=4.2s^{-1}mM^{-1}$, $r_{2}=6.7s^{-1}mM^{-1}$ are the Gadolinium-based contrast agent relaxivities; TR=8.24 ms and TE=3.1 ms are the repetition and echo times; $\theta_{FA}=12^{\circ }$ the flip angle; and T1[t] and T2[t] the longitudinal and transversal relaxation times at time *t*, estimated as described in [9]. The relaxation time is assumed to decrease with contrast agent concentration and relaxivities, $1/Ti\left[t\right]=1/Ti\left[0\right]+r_{i}\cdot c\left[t\right],\;i=1,2$. After obtaining the curves, we estimated the area under the concentration-time curve,
(4)$$AUC_{Con}=\sum_{t=0}^{T-1}c\left[t\right],$$
and its slope,
(5)$$Slope_{Con}=\frac{\left(T-t^{*}\right)\cdot \sum_{t=t^{*}}^{T-1}t\cdot c[t]-\sum_{t=t^{*}}^{T-1}t\cdot \sum_{t=t^{*}}^{T-1}c[t]}{\left(T-t^{*}\right)\cdot \sum_{t=t^{*}}^{T-1}t^{2}-{\left(\sum_{t=t^{*}}^{T-1}t\right)}^{2}}.$$

#### 2.3.3. Methods to Analyse Changes in the Radial Power Spectrum

The analysis of the radial power spectrum (RPS) permits scrutinising alterations in the spatial frequency domain due to the intravenous injection of the contrast agent [17]. Similar to previous works in the field [7,8], we computed the RPS for each region of interest and each time point after the peak of bolus arrival by calculating the magnitude spectra and averaging it over all frequencies in concentric rings of width one, as follows
(6)$$R\left[s;t\right]=\frac{1}{K}\sum_{k=1}^{K}\frac{1}{2\pi }\int_{0}^{2\pi }|F_{k}\left(s\;\cos\left(\theta \right),s\;\sin\left(\theta \right),t\right)|\;d\theta ,$$
where $F_{k}$ denotes the 2D discrete Fourier transform of the *k*-th axial slice of the input volume, K=42 the number of slices, and $s=\mathrm{round}\left(\sqrt{u^{2}+v^{2}}\right)$ and $\theta =\tan^{-1}\left(v/u\right)$ are polar coordinates.

To analyse these signals that include 201 patients × 129 rings × 17 time points, we opted for reducing their cardinality to only a set of measurements per patient. We achieved this by reducing it in the direction of the rings and time using the multivariate functional principal component analysis proposed by Happ & Greven [18]. Let $R_{p}\left[s;t\right]$ be the RPS over time of patient p∈[1,201], s∈[0,128], t∈[4,20], the overall process consists of four steps. First, we centred each variable by subtracting its mean value across patients. Second, we computed *E* eigenfunctions $\Phi_{k}\left[s;t\right]$ and corresponding scores $\xi_{pk}\left[s\right]$ by maximising $\sum_{p}{\xi_{pk}\left[s\right]}^{2}$, where
(7)$$\xi_{pk}\left[s\right]=\sum_{t=t^{*}}^{T-1}\Phi_{k}\left[s;t\right]\cdot R_{p}\left[s;t\right],$$
subject to $\left|\right|\Phi_{k}\left[s\right]{\left|\right|}^{2}=1$. We set E=5 as the resulting eigenfunctions explained 99% of the data variance. Third, all of these scores $\xi_{pk}\left[s\right]$ were arranged in a matrix form, $\Xi \in R^{201\times 129\times E}$, such that the $p^{th}$ row contained $\left(\xi_{p1}\left[0\right],\ldots ,\xi_{p1}\left[128\right],\ldots ,\xi_{p5}\left[0\right],\ldots ,\xi_{p5}\left[128\right]\right)$. Fourth, we calculated principal component scores by means of eigenanalysis on the covariance matrix of Ξ. We considered only the first mode of variation since it explained around the 98% of the data variance.

### 2.4. Validation against Clinical Parameters

We considered one-way analysis of variance (ANOVA) to evaluate the considered measures of contrast uptake/washout against clinical visual assessments related to SVD severity (i.e., Fazekas and total SVD scores), and multiple linear regression to establish whether age, diabetes, stroke lesion subtype, WMH volume, and stroke lesion volume were associated with them, after adjusting for biological sex, mean arterial blood pressure, hyperlipidaemia, and smoking status. For ANOVA tests, we used the following notation: (F(k−1,n−k)=F-value, *p*, $\eta^{2}$, $\omega^{2}$), where *k* represents the number of groups (i.e., 7 and 5 for Fazekas and total SVD score, respectively), *n* the sample size, F-value the F test statistic, *p* the *p*-value, and $\eta^{2}$ and $\omega^{2}$ are effect size estimators which indicate the proportion of data variance explained by predictors. For multiple linear regression, we reported β (95% CI) and *p*-value for each predictor, which indicate their weight in the model and whether the predictor is significant in the model, respectively. Also, we reported the adjusted $R^{2}$ values which specify the percentage of data variance in the outcome variable explained by predictors. We carried out our statistical analyses using RStudio v1.1.456 with R v3.5.1.

## 3. Results

### 3.1. Comparison of Effect Sizes

We evaluated the effect of the measures of contrast uptake/washout computed in cerebrospinal fluid and WMH against two relevant visual clinical ratings (Fazekas and SVD scores).

The effect of the burden of WMH on most vascular function measures computed from the cerebrospinal fluid regions was significant ($AUC_{Con}$: F(6,190)=2.54, p<0.05, $\eta^{2}=7\%$, $\omega^{2}=5\%$; $Slope_{Enh}$: F(6,190)=2.87,p<0.05, $\eta^{2}=8\%$, $\omega^{2}=5\%$; $Slope_{Con}$: F(6,190)=3.71, p<0.01, $\eta^{2}=11\%$, $\omega^{2}=8\%$, RPS: F(6,190)=2.40, p<0.05, $\eta^{2}=7\%$, $\omega^{2}=4\%$), except on the $AUC_{Enh}$ ($AUC_{Enh}$: F(6,190)=0.55, p>0.10, $\eta^{2}=2\%$, $\omega^{2}=-1\%$), as shown in Figure 3. The higher the Fazekas score, the lower the AUCs and the rising the Slopes. The association was stronger (higher $\eta^{2}$ and $\omega^{2}$) for the $Slope_{Con}$ than the other four measures. The total burden of neuroimaging features of SVD, given by the total SVD score, was associated with the $AUC_{Con}$, $Slope_{Enh}$, and $Slope_{Con}$ ($AUC_{Con}$: F(4,192)=2.42, p=0.05, $\eta^{2}=5\%$, $\omega^{2}=3\%$; $Slope_{Enh}$: F(4,192)=2.37, p=0.05, $\eta^{2}=5\%$, $\omega^{2}=3\%$; $Slope_{Con}$: F(4,192)=3.43, p<0.01, $\eta^{2}=7\%$, $\omega^{2}=5\%$), but not with the other measurements ($AUC_{Enh}$: F(4,192)=0.48, p>0.1, $\eta^{2}=2\%$, $\omega^{2}=0\%$; RPS: F(4,192)=2.17, p>0.05, $\eta^{2}=4\%$, $\omega^{2}=2\%$). The higher the total SVD score, the lower the AUCs and RPS, the rising the Slopes.

The impact of the burden of WMH on most measures of contrast uptake/washout from WMH was significant ($AUC_{Enh}$: F(6,190)=2.61, p<0.05, $\eta^{2}=8\%$, $\omega^{2}=5\%$; $AUC_{Con}$: F(6,190)=4.35, p<0.001, $\eta^{2}=12\%$, $\omega^{2}=9\%$; $Slope_{Enh}$: F(6,190)=2.93,p<0.01, $\eta^{2}=9\%$, $\omega^{2}=6\%$; $Slope_{Con}$: F(6,190)=2.73, p<0.05, $\eta^{2}=8\%$, $\omega^{2}=5\%$), except on the RPS (RPS: F(6,190)=1.48, p>0.10, $\eta^{2}=5\%$, $\omega^{2}=1\%$), as depicted in Figure 4. The higher the Fazekas score, the lower the AUCs and the rising the Slopes. The association was stronger for the $AUC_{Con}$ than the other four measures. The burden SVD features was only associated with the $AUC_{Con}$ (F(4,192)=3.09, p<0.05, $\eta^{2}=6\%$, $\omega^{2}=4\%$), but not with the other measurements ($AUC_{Enh}$: F(4,192)=1.86, p>0.1, $\eta^{2}=4\%$, $\omega^{2}=2\%$; $Slope_{Enh}$: F(4,192)=1.71, p>0.10, $\eta^{2}=3\%$, $\omega^{2}=1\%$; $Slope_{Con}$: F(4,192)=2.12, p>0.05, $\eta^{2}=4\%$, $\omega^{2}=2\%$; RPS: F(4,192)=2.01, p>0.05, $\eta^{2}=4\%$, $\omega^{2}=2\%$). The higher the total SVD score, the lower the AUCs, the rising the Slopes.

### 3.2. Relationship between Contrast Uptake/Washout Measures and Clinical Variables

We carried out multiple linear regression to investigate whether age, diabetes, stroke lesion subtype, WMH volume, and stroke lesion volume were associated with semiquantitative measures of contrast uptake/washout, after adjusting for biological sex, mean arterial pressure, smoker, and hyperlipidaemia. Corresponding regression results are condensed in Table 1 and Table A1.

In the cerebrospinal fluid region, four vascular function measures showed associations with clinical variables (p<0.001), RPS did not (p>0.1). The associations were stronger (lower *p*-value, higher coefficients of determination) when considering concentration-time curves instead of enhancement-time curves (AUC: $R_{Enh}^{2}=16\%$ vs. $R_{Con}^{2}=36\%$; Slope: $R_{Enh}^{2}=10\%$ vs. $R_{Con}^{2}=33\%$). Age was negatively associated the AUCs and Slopes in the CSF ($AUC_{Enh}:\beta =-4.27\times 10^{-2}\;[95\%\mathrm{CI}-5.89\times 10^{-2},\;-2.65\times 10^{-2}],\;p<0.001$; $AUC_{Con}:\beta =-1.52\times 10^{-2}\;[95\%\mathrm{CI}-1.87\times 10^{-2},\;-1.17\times 10^{-2}],\;p<0.001$; $Slope_{Enh}:\beta =-3.51\times 10^{-5}\;[95\%\mathrm{CI}-5.27\times 10^{-5},\;-1.76\times 10^{-5}],\;p<0.001$; $Slope_{Con}:\beta =-1.29\times 10^{-5}\;[95\%\mathrm{CI}-1.61\times 10^{-5},\;-9.66\times 10^{-6}],\;p<0.001$). Clinical parameters predicted most variance in $AUC_{Con}$ values compared to the four other methods (Adjusted *R*${}^{2}:AUC_{Con}=36\%$ vs. $Slope_{Con}=33,RPS=2\%,AUC_{Enh}=16\%,Slope_{Enh}=10\%$).

In the deep grey matter, $AUC_{Con}$, $Slope_{Con}$, and RPS showed associations with clinical variables. Age was a strong predictor was negatively associated with $AUC_{Con}$ and $Slope_{Con}$ ($AUC_{Con}:\beta =-2.29\times 10^{-3}\;[95\%\mathrm{CI}-3.29\times 10^{-3},\;-1.30\times 10^{-3}],\;p<0.001$; $Slope_{Con}:\beta =-1.35\times 10^{-6}\;[95\%\mathrm{CI}-2.43\times 10^{-6},\;-2.59\times 10^{-7}],\;p<0.05$). A diagnosis of diabetes was associated with an increase in the $AUC_{Con}$ ($AUC_{Con}:\beta =5.37\times 10^{-2}\;[95\%\mathrm{CI}2.28\times 10^{-2},\;8.46\times 10^{-2}],\;p<0.001$). WMH volume was negatively associated with the RPS ($RPS:\beta =-3.19\times 10^{3}\;[95\%\mathrm{CI}-5.51\times 10^{3},\;-8.81\times 10^{3}],\;p<0.01$). Clinical parameters predicted the variance in $AUC_{Con}$ values the best compared to other methods (Adjusted *R*${}^{2}:AUC_{Con}=13\%$ vs. $Slope_{Con}=8,RPS=5\%,AUC_{Enh}=3\%,Slope_{Enh}=3\%$).

In normal-appearing white matter, clinical parameters were weakly or not associated with the five measures of vascular function (i.e. 0.01<p<0.05 and p>0.1, respectively) as they explained between 1% and 6% of their variability. Age predicted the $AUC_{Con}$ significantly ($AUC_{Con}:\beta =-1.81\times 10^{-3}\;[95\%\mathrm{CI}-2.70\times 10^{-3},\;-9.08\times 10^{-4}],\;p<0.001$).

In WMH, clinical variables predicted 13% and 25% of the variance in $AUC_{Enh}$ and $AUC_{Con}$ (p<0.001). Age, diabetes, and WMH volume were significantly and consistently associated with $AUC_{Enh}$ and $AUC_{Con}$ (p<0.05 and p<0.01, respectively). An increase in $AUC_{Enh}$ and $AUC_{Con}$ was associated with younger age ($AUC_{Enh}:\beta =-6.81\times 10^{-3}\;[95\%\mathrm{CI}-1.26\times 10^{-2},\;-1.07\times 10^{-3}],\;p<0.05$; $AUC_{Con}:\beta =-2.88\times 10^{-3}\;[95\%\mathrm{CI}-4.07\times 10^{-3},\;-1.69\times 10^{-3}],\;p<0.001$), diabetes diagnosis ($AUC_{Enh}:\beta =2.72\times 10^{-1}\;[95\%\mathrm{CI}9.31\times 10^{-1},\;4.51\times 10^{-1}],\;p<0.01$; $AUC_{Con}:\beta =6.13\times 10^{-2}\;[95\%\mathrm{CI}2.44\times 10^{-2},\;9.82\times 10^{-2}],\;p<0.01$), and less WMH volume ($AUC_{Enh}:\beta =-4.46\;[95\%\mathrm{CI}-7.17,\;-1.75],\;p<0.01$; $AUC_{Con}:\beta =-1.08\;[95\%\mathrm{CI}-1.63,\;-0.52],\;p<0.001$). Moreover, the stroke lesion volume was positively associated with the $AUC_{Enh}$ ($AUC_{Enh}:\beta =6.38\;[95\%\mathrm{CI}1.05,\;11.7],\;p<0.01$).

In stroke lesions, all vascular function measures were associated with clinical variables (p<0.05). The associations between clinical parameters and $AUC_{Enh}$ and $AUC_{Con}$ were clearer compared to those of the rest ($28\%\leq R^{2}\leq 33\%$ vs. $4\%\leq R^{2}\leq 14\%$). Three variables strongly predicted the outcome variables for some of these models: diabetes, stroke lesion subtype, and stroke lesion volume. A diagnosis of diabetes was significantly associated with an increase in AUCs and Slopes ($AUC_{Enh}:\beta =1.13\;[95\%\mathrm{CI}0.61,\;1.65],\;p\;<\;0.001$; $AUC_{Con}:\beta =0.31\;[95\%\mathrm{CI}0.20,\;0.41],\;p<0.001$; $Slope_{Enh}:\beta =6.85\times 10^{-4}\;[95\%\mathrm{CI}-1.45\times 10^{-6},\;1.37\times 10^{-3}],\;p=0.05$; $Slope_{Con}:\beta =2.60\times 10^{-4}\;[95\%\mathrm{CI}1.04\times 10^{-4},\;4.16\times 10^{-4}],\;p<0.01$), but not with RPS (p>0.1). Cortical strokes had higher AUCs and $Slope_{Con}$ ($AUC_{Enh}:\beta =-0.83\;[95\%\mathrm{CI}-1.20,\;-0.47],\;p<0.001$; $AUC_{Con}:\beta =-0.12\;[95\%\mathrm{CI}-0.19,\;-0.05],\;p<0.01$; $Slope_{Con}:\beta =-1.29\times 10^{-4}\;[95\%\mathrm{CI}-2.38\times 10^{-4},\;-1.92\times 10^{-5}],\;p<0.05$). Stroke volume was associated with an increase in $AUC_{Enh}$, $Slope_{Enh}$, and RPS ($AUC_{Enh}:\beta =2.41\times 10^{1}\;[95\%\mathrm{CI}1.01\times 10^{1},\;3.82\times 10^{1}],\;p<0.001$; $Slope_{Enh}:\beta =3.37\times 10^{-2}\;[95\%\mathrm{CI}1.52\times 10^{-2},\;5.22\times 10^{-2}],\;p<0.001$; $RPS:\beta =7.16\times 10^{3}\;[95\%\mathrm{CI}2.82\times 10^{3},\;1.15\times 10^{4}],\;p<0.01$).

## 4. Discussion

In this work, we compared the performance of five semiquantitative methods for analysing signal-time trajectories of Gadolinium-based contrast agent in reflecting small vessel disease burden within healthy and pathological intracranial brain regions. The five methods estimate and analyse the slopes and area under the enhancement-time and concentration-time curves and changes in the power spectrum of the contrast-enhancement signal over time.

The considered semiquantitative measurements assessing contrast uptake/washout provide different yet complementary information related to cerebrovascular dysfunction. First, the areas under the enhancement-time/concentration-time curves describe two processes that may cause signal change in tissue jointly: accumulation of contrast agent in the extravascular extracellular space due to blood–brain barrier leakage and total volume of blood. Since the former effect is expected to be subtle in small vessel disease, we expect these areas under the curves to reflect more total blood volume. In regions filled with cerebrospinal fluid, increases in areas under the curves could be caused by contrast agent leakage due to an impaired blood-cerebrospinal fluid barrier [1,2,3]. Second, the slopes of the enhancement-time/concentration-time curves describe the rate at which the contrast agent washes out of brain tissues. While a positive slope reflects uptake of contrast agent in tissue over time, a negative slope reflects contrast washout over time. The magnitude of such a change indicates the speed at which it happens: the higher the magnitude, the faster the change over time. Thus, the slower the washout rate, the more the contrast agent stays in tissue potentially due to its accumulation in the extracellular extravascular space. In regions filled with cerebrospinal fluid, a slope different from zero may reveal impairment of the blood–CSF barrier. Third, the analysis of the radial power spectrum permits quantifying changes in frequencies over time cohort-wise [8]. A positive or negative value expresses distancing from the mean behaviour (described by each eigenfunction) cohort-wise.

We performed a one-way ANOVA to determine the effect of the burden of WMH and neuroimaging features of SVD, expressed in terms of the Fazekas and total SVD scores, on the five measures of contrast uptake/washout from CSF and WMH. Most effects were significant on measurements extracted from both CSF and WMH considering Fazekas scores. The only significant effects when considering the total SVD score were the slope of the concentration-time curve extracted from regions filled with CSF and the area under the concentration-time extracted from WMH. These results imply that most measurements vary depending on the overall burden of WMH in the brain, but only concentration-time measurements capture additional aspects of vascular dysfunction on univariate analyses. Moreover, the relationship between measurements and clinical visual scores was more evident (lower *p*-values) when extracted from concentration-time curves. Therefore, the analysis of concentration-time curves is more reliable than enhancement-time curves since the former includes adjustment for contrast agent relaxivities, imaging parameters, and relaxation times of each region of interest. The slopes and areas under the curves exhibited opposite trends: slopes increased with WMH and total SVD score, whilst areas decreased. Examination on larger datasets with more varied states of brain vascular and other pathologies is needed to more fully understand the factors influencing the measurement of these potentially valuable differential vascular dysfunction parameters.

We performed multiple linear regression to establish whether clinical parameters (age, biological sex, mean arterial pressure, hyperlipidaemia, smoker, diabetes, stroke lesion subtype, WMH and stroke lesion volume) determined the extent of enhancement in cerebrospinal fluid, deep grey matter, normal-appearing white matter, WMH, and stroke lesion. The strength of the associations were consistently higher when considering measurements from the concentration-time curves and not from the enhancement-time curves consistent with the literature [1,4,5,6,19,20,21,22,23]. This might imply that the use of imaging parameters to obtain these former trajectories provide better estimates of the contrast uptake/washout. While the analysis of the area under the concentration-time curve was explained the best by clinical parameters regardless of the region of interest, the analysis of the radial power spectrum did not identify associations as they were the weakest compared to the other four measurements. The presence of noise and signal drift in similar levels to the signal changes has been acknowledged previously [9,24]. The RPS, reflective of the cumulative effect of the whole frequency spectrum forming the signal, is a sensitive measure worth further evaluation for this purpose once appropriate noise filtering procedures have been applied.

Estimates of contrast agent update/washout in cerebrospinal fluid-filled spaces, deep grey matter, normal-appearing white matter, and WMH were negatively associated with age. Given that the total volume of blood decreases with age [25] and that leakage in small vessel disease is expected to be subtle [3], this outcome suggests that the enhancement in the capillaries might be overshadowing the enhancement due to leakage and, hence, the semiquantitative methods considered in this work examine vascular surface area, in accordance with previous research in the field [26]. In WMH regions, our results indicate that their enhancement decreases with the extent of demyelination and axonal damage [27], consistent with previous findings [28,29,30]. In deep grey matter, WMH, and stroke lesions, diabetes influenced the contrast uptake/washout estimates: higher values in diabetic vs. non-diabetic patients. This outcome suggests that diabetic patients may present a reduction in capillary density or a higher impairment of the blood–brain barrier. In both cases, this relationship agrees with previous studies in which both hyper- and hypo-glycemia have been associated with cerebrovascular alterations [31] and compromised the blood–brain barrier [32,33]. In stroke lesions, large cortical strokes exhibited the highest contrast uptake/washout estimates, much more evident in diabetic patients. Further research in these directions is needed to account for the interaction between the capillaries and the extracellular extravascular space and their contribution to the overall enhancement.

Future work should consider comparing more semiquantitative and quantitative approaches for analysing concentration-time curves and assessing their robustness against imaging artefacts as they compromise current assessments [3,9,10,11].

## Figures and Tables

**Figure 1 jimaging-06-00043-f001:**
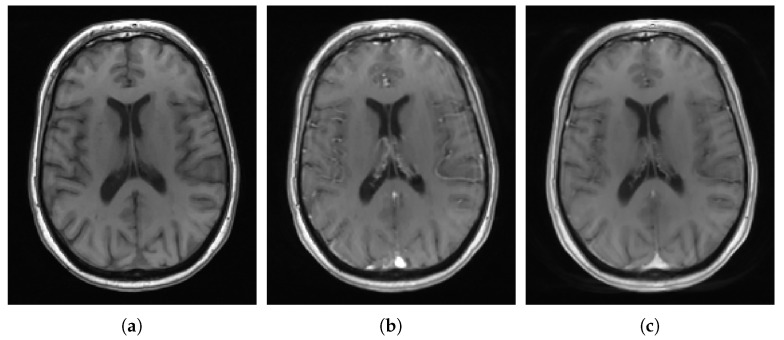
Dynamic contrast-enhanced magnetic resonance imaging acquisition. From left to right, axial slices of T1-w scans before, 1 min after, and 24 min after intravenous Gadolinium-based contrast agent injection, respectively. (**a**) Before contrast, (**b**) 1 min after, (**c**) 24 min after.

**Figure 2 jimaging-06-00043-f002:**
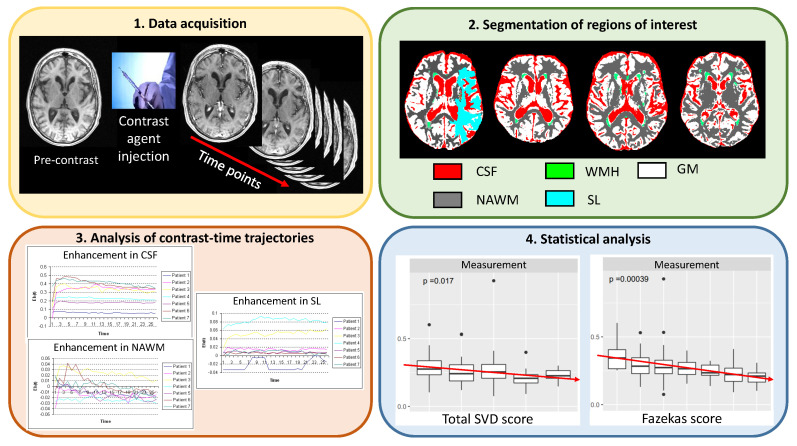
High level schematic of our processing pipeline. First, we acquire dynamic contrast-enhanced scans for each patient. Second, we segment regions of interest into cerebrospinal fluid, grey matter, normal-appearing white matter, white matter hyperintensities, and stroke lesions. Third, we analyse contrast-time trajectories using different approaches: slopes and areas under the enhancement-time and concentration-time curves and changes over time in the radial power spectrum. Fourth, we use ANOVA and multiple linear regression to examine the effect of the burden of white matter hyperintensities (Fazekas score) and all neuroimaging features of small vessel disease (Total SVD score). In Step 4, data points located outside of the whiskers of each boxplot correspond to outliers. CSF: cerebrospinal fluid. WMH: white matter hyperintensity. NAWM: normal-appearing white matter. GM: grey matter. SL: stroke lesion.

**Figure 3 jimaging-06-00043-f003:**
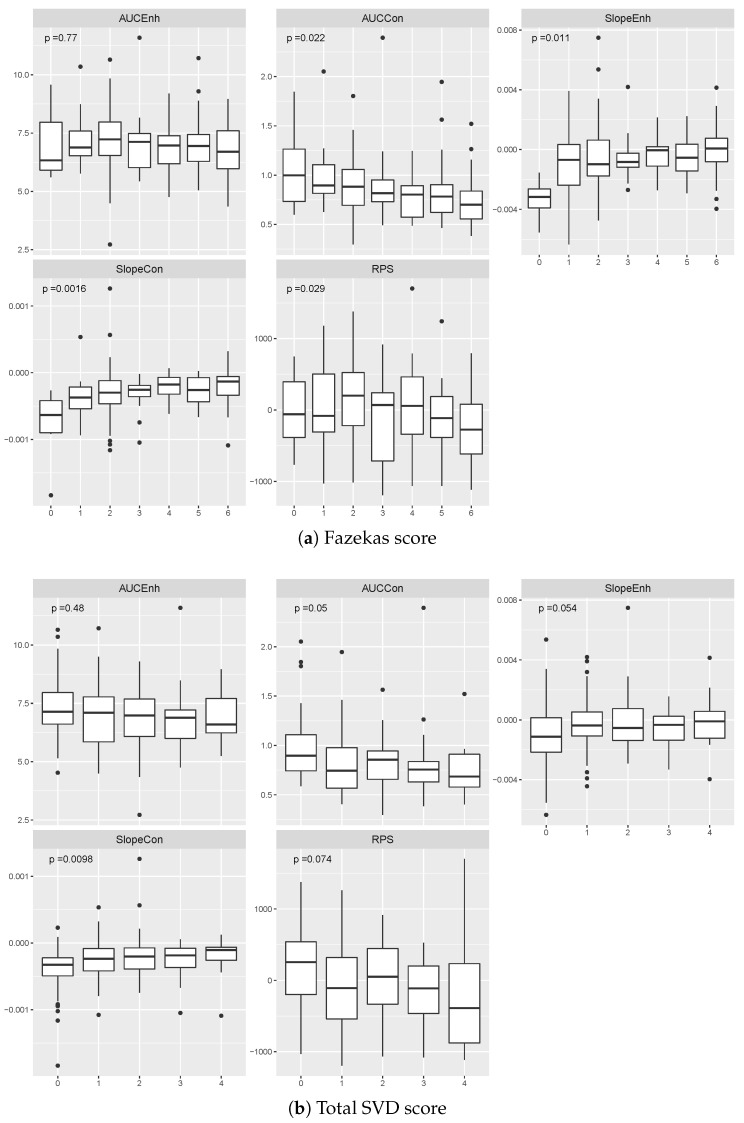
Estimated contrast uptake/washout extracted from cerebrospinal fluid for patients grouped by their (**a**) Fazekas and (**b**) total SVD scores. We computed the p-values using the ANOVA test. Data points located outside of the whiskers of each boxplot correspond to outliers.

**Figure 4 jimaging-06-00043-f004:**
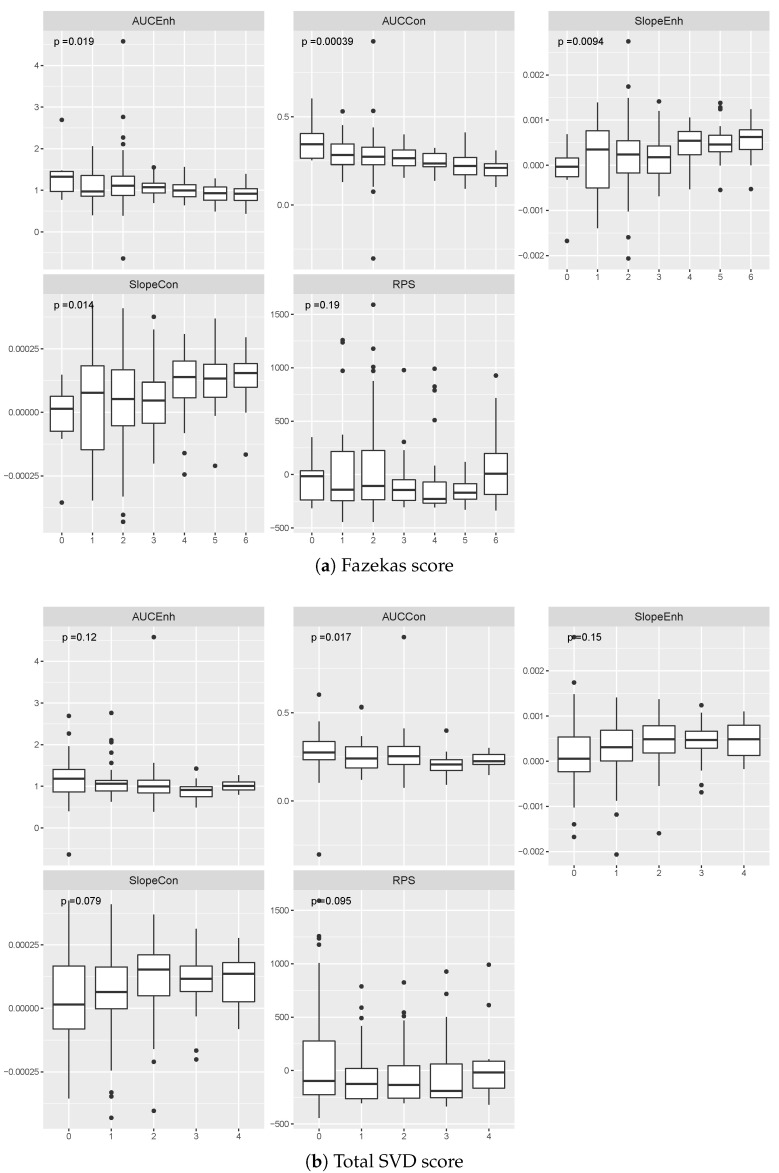
Estimated contrast uptake/washout extracted from white matter hyperintensities for patients grouped by their (**a**) Fazekas and (**b**) total SVD scores. We computed the p-values using the ANOVA test. Data points located outside of the whiskers of each boxplot correspond to outliers.

**Table 1 jimaging-06-00043-t001:** Adjusted *R*${}^{2}$ and *p*-values of multiple linear regression with semiquantitative contrast uptake/washout estimates per region of interest as predicted variables and clinical parameters as predictors. This table is a simplified version of Table A1. Adjusted *R*${}^{2}$ values are shown as percentages. Significant associations appear in bold. Stroke type: 0—cortical, 1—lacunar. ROI: region of interest. CI: confidence interval. CSF: cerebrospinal fluid. DGM: deep grey matter. NAWM: normal-appearing white matter. WMH: white matter hyperintensity. SL: stroke lesion. RPS: radial power spectrum.

						Stroke	WMH	SL
ROI	Method	$R^{2}$	*p*-Value	Age	Diabetes	Type	Volume	Volume
CSF	$AUC_{Enh}$	16	**2.17**×$10^{-6}$	**5.02 ×$10^{-7}$**	5.85 ×$10^{-1}$	7.88×$10^{-1}$	3.90 ×$10^{-1}$	1.61 ×$10^{-1}$
	$AUC_{Con}$	36	**4.44 ×$10^{-16}$**	**3.77 ×$10^{-15}$**	8.32 ×$10^{-1}$	7.35 ×$10^{-1}$	7.69 ×$10^{-1}$	9.61 ×$10^{-1}$
	$Slope_{Enh}$	10	**5.92 ×$10^{-4}$**	**1.12 ×$10^{-4}$**	1.56 ×$10^{-1}$	6.24 ×$10^{-1}$	1.55 ×$10^{-1}$	4.43 ×$10^{-1}$
	$Slope_{Con}$	33	**9.97 ×$10^{-15}$**	**2.46 ×$10^{-13}$**	2.19 ×$10^{-1}$	3.98 ×$10^{-1}$	4.76 ×$10^{-1}$	2.03 ×$10^{-1}$
	RPS	2	1.78 ×$10^{-1}$	8.97 ×$10^{-1}$	7.46 ×$10^{-1}$	7.10 ×$10^{-1}$	6.79 ×$10^{-2}$	3.64 ×$10^{-1}$
DGM	$AUC_{Enh}$	3	8.11 ×$10^{-2}$	2.97 ×$10^{-1}$	**2.75 ×$10^{-2}$**	4.51 ×$10^{-1}$	8.53 ×$10^{-2}$	1.12 ×$10^{-1}$
	$AUC_{Con}$	13	**6.20 ×$10^{-5}$**	**9.30 ×$10^{-6}$**	**7.47 ×$10^{-4}$**	6.04 ×$10^{-1}$	7.75 ×$10^{-1}$	8.18 ×$10^{-1}$
	$Slope_{Enh}$	3	9.77 ×$10^{-2}$	1.93 ×$10^{-1}$	6.31 ×$10^{-1}$	6.54 ×$10^{-1}$	3.97 ×$10^{-1}$	6.90 ×$10^{-1}$
	$Slope_{Con}$	8	**3.59 ×$10^{-3}$**	**1.55 ×$10^{-2}$**	3.23 ×$10^{-1}$	7.59 ×$10^{-1}$	1.20 ×$10^{-1}$	9.86 ×$10^{-1}$
	RPS	5	**3.33 ×$10^{-2}$**	4.89 ×$10^{-1}$	1.09 ×$10^{-1}$	8.24 ×$10^{-1}$	**7.05 ×$10^{-3}$**	4.57 ×$10^{-1}$
NAWM	$AUC_{Enh}$	3	1.27 ×$10^{-1}$	2.68 ×$10^{-1}$	6.64 ×$10^{-1}$	4.53 ×$10^{-1}$	**2.47 ×$10^{-2}$**	5.82 ×$10^{-1}$
	$AUC_{Con}$	6	**1.07 ×$10^{-2}$**	**1.03 ×$10^{-4}$**	1.83 ×$10^{-1}$	3.84 ×$10^{-1}$	6.17 ×$10^{-1}$	3.19 ×$10^{-1}$
	$Slope_{Enh}$	−1	5.94 ×$10^{-1}$	7.04 ×$10^{-1}$	9.14 ×$10^{-1}$	2.27 ×$10^{-1}$	2.55 ×$10^{-1}$	3.50 ×$10^{-1}$
	$Slope_{Con}$	2	1.43 ×$10^{-1}$	2.73 ×$10^{-1}$	5.44 ×$10^{-1}$	1.94 ×$10^{-1}$	1.08 ×$10^{-1}$	1.09 ×$10^{-1}$
	RPS	−2	8.53 ×$10^{-1}$	9.57 ×$10^{-1}$	8.11 ×$10^{-1}$	5.49 ×$10^{-1}$	8.13 ×$10^{-1}$	9.97 ×$10^{-1}$
WMH	$AUC_{Enh}$	13	**3.60 ×$10^{-5}$**	**2.04 ×$10^{-2}$**	**3.07 ×$10^{-3}$**	7.86 ×$10^{-1}$	**1.37 ×$10^{-3}$**	**1.93 ×$10^{-2}$**
	$AUC_{Con}$	25	**2.49 ×$10^{-10}$**	**3.38 ×$10^{-6}$**	**1.25 ×$10^{-3}$**	9.94 ×$10^{-1}$	**1.98 ×$10^{-4}$**	8.61 ×$10^{-1}$
	$Slope_{Enh}$	4	6.11 ×$10^{-2}$	4.59 ×$10^{-1}$	5.35 ×$10^{-1}$	2.79 ×$10^{-1}$	4.04 ×$10^{-1}$	1.21 ×$10^{-1}$
	$Slope_{Con}$	4	7.23 ×$10^{-2}$	6.88 ×$10^{-2}$	2.92 ×$10^{-1}$	3.00 ×$10^{-1}$	3.25 ×$10^{-1}$	7.95 ×$10^{-1}$
	RPS	−3	9.58 ×$10^{-1}$	4.82 ×$10^{-1}$	3.88 ×$10^{-1}$	4.72 ×$10^{-1}$	8.78 ×$10^{-1}$	6.12 ×$10^{-1}$
SL	$AUC_{Enh}$	33	**9.90 ×$10^{-10}$**	5.04 ×$10^{-1}$	**3.43 ×$10^{-5}$**	**1.52 ×$10^{-5}$**	1.07 ×$10^{-1}$	**8.69 ×$10^{-4}$**
	$AUC_{Con}$	28	**6.33 ×$10^{-8}$**	2.21 ×$10^{-1}$	**6.73 ×$10^{-8}$**	**1.92 ×$10^{-3}$**	7.88 ×$10^{-2}$	6.87 ×$10^{-2}$
	$Slope_{Enh}$	14	**6.30 ×$10^{-4}$**	6.69 ×$10^{-1}$	**5.05 ×$10^{-2}$**	6.02 ×$10^{-2}$	1.63 ×$10^{-1}$	**4.36 ×$10^{-4}$**
	$Slope_{Con}$	13	**1.02 ×$10^{-3}$**	3.41 ×$10^{-1}$	**1.28 ×$10^{-3}$**	**2.17 ×$10^{-2}$**	1.04 ×$10^{-1}$	7.06 ×$10^{-2}$
	RPS	4	**4.51 ×$10^{-2}$**	9.52 ×$10^{-1}$	6.26 ×$10^{-1}$	6.31 ×$10^{-1}$	**2.95 ×$10^{-2}$**	**1.34 ×$10^{-3}$**

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
