# Peer review of "Examining the Relationship between Semiquantitative Methods Analysing Concentration-Time and Enhancement-Time Curves from Dynamic-Contrast Enhanced Magnetic Resonance Imaging and Cerebrovascular Dysfunction in Small Vessel Disease†"

_2313-433X, 2020, doi:10.3390/jimaging6060043_

Round 1
Reviewer 1 Report
This paper is to calculate the slopes and areas over enhancement- and concentration-time curves and changes in the power spectrum to study 5 brain tissues using DEC-MRI scans from the patients of small vessel disease presenting mild stroke. This paper is well-written to include meaningful results through statistical analysis. I have two comments.
- The title seems to focus on semiquantitative methods so that readers may expect a detailed description of technical methods. However, the paper didn’t describe the details of the methods. Therefore, the title may be revised to represent the main results of best representing, concentration-time curve.
- The last paragraph in the Introduction may be improved to emphasize the results rather than compare the 5 techniques.
Author Response
Many thanks for your feedback. Please see the attachment.

Reviewer 2 Report
This article tests 5 different analytical methods on DCE-MRI images from a cohort of 201 patients about 1 month after a stroke episode. The goal is to find the best way to extract information about cerebrovascular dysfunctions.
In order to do this, processed data are related to non parametric clinical scores which refer to gravity of white matter hyperintensities or to presence of alterations found in small vessel disease. In addition, the authors tried to work out how processed DCE-MRI data correlated with a set of clinical findings after adjustments for patient charcteristics: sex, pressure, smoke habit, hyperlipidaemia.
The main finding is that taking in account contrast agent relaxivities, imaging parameters and relaxation times of each region of interests increases the association of DCE-MRI data to non parametric clinical scores. Similarly, results from data processed accordingly showed an higher association with clinical parameters.
I have no specific problem with the work, up to discussion. Where I object is about the extrapolation of conclusions, especially because authors wish to read the results obtained in terms of vascular density and endothelial leakage. In this optic, standing to text, they seem to claim a link between an analytical method and a biological characteristic (vascular density or endothelial leakage) only on the basis of the changes expected for an average patient for that same biological parameter.
The study is not planned to test for these hypotheses. Similarly, the paper is not written in order to discuss this conclusion. The authors seem to ignore that the correlation they observed between data and clinical findings explains, at best, only 1/3 of the observed variance. Therefore association between a reduced analytical value and a reduced parameter (e.g.: vascular density) can be completely unrelated even if expected from a logic standpoint. Furthermore, the paper does not present other independent estimation of vascular densities and vascular leakage.
In addition, the study here presented was carried out through a single sampling point after stabilization of the vascular situation and cannot considered planned to test for association between DCE-MRI images and controlled conditions of increased or reduced vascular density or BBB alterations. In these respect the same authors suggest to expand and diversify samples in terms of different pathologies, especially because imaging artefacts compromise current assessments (last sentence).
As a further point, it must also be considered that area under curve (AUC) and slope are related entities. The first is the integral of slope as a function of time. Assuming that vascular density is linked to AUC and slope to BBB integrity would lead to the conclusion that vascular density and BBB integrity are somehow related.
In conclusion I think that Abstract and Discussion sections must be rewritten and should stick to the findings the paper has highlighted.
Minor points:
- the related analytical methods used for elaboration of DCE-MRI data are referred as techniques at pag. 2 and in the discussion. This mislead the reader to look for different acquisition approaches more than to look for different elaborations of the same imaging data.
- the associative descriptors you used (eta quadro and w2) were not described in the methods section and therefore are meaningless for the general reader.
- Figures 2 and 3. Please report in the legend the meaning of the black points.
Author Response

(The authors gave the same response as above.)

Round 2
Reviewer 2 Report
Authors positively addressed all the expressed concerns.